# Psychogenic Nonepileptic Seizures—High Mortality Rate Is a ‘Wake-Up Call’

**DOI:** 10.3390/jpm13060892

**Published:** 2023-05-25

**Authors:** Catherine A. Carlson

**Affiliations:** Minnesota Judicial Branch Psychological Services Division, Minneapolis, MN 55487, USA; carlsonc32@gmail.com

**Keywords:** epilepsy, psychogenic nonepileptic seizures, vEEG, conversion disorder, drug-resistant epilepsy, scalp EEG-negative epileptic seizure, epilepsy monitoring units, anti-epileptic drugs, intracranial monitoring

## Abstract

Patients with epilepsy have an elevated mortality rate compared to the general population and now studies are showing a comparable death ratio in patients diagnosed with psychogenic nonepileptic seizures. The latter is a top differential diagnosis for epilepsy and the unexpected mortality rate in these patients underscores the importance of an accurate diagnosis. Experts have called for more studies to elucidate this finding but the explanation is already available, embedded in the existing data. To illustrate, a review of the diagnostic practice in epilepsy monitoring units, of the studies examining mortality in PNES and epilepsy patients, and of the general clinical literature on the two populations was conducted. The analysis reveals that the scalp EEG test result, which distinguishes a psychogenic from an epileptic seizure, is highly fallible; that the clinical profiles of the PNES and epilepsy patient populations are virtually identical; and that both are dying of natural and non-natural causes including sudden unexpected death associated with confirmed or suspected seizure activity. The recent data showing a similar mortality rate simply constitutes more confirmatory evidence that the PNES population consists largely of patients with drug-resistant scalp EEG-negative epileptic seizures. To reduce the morbidity and mortality in these patients, they must be given access to treatments for epilepsy.

## 1. Introduction

After epilepsy, psychogenic nonepileptic seizures (PNES) are likely the second most common diagnosis made by epileptologists [1]. PNES are defined as paroxysmal episodes that clinically resemble epileptic seizures but unlike the latter, do not show an epileptiform discharge on the surface electrodes of a video electroencephalogram (vEEG) [2]. The absence of an epileptiform discharge is considered proof that the seizure is not epileptic, and thus it presumably has a psychological origin [2]. In modern nomenclature, PNES warrants a diagnosis of Conversion Disorder [3]. It has been postulated that PNES are essentially dissociations that operate as a defensive psychological mechanism that use the mind as a defense to deal with trauma [4]. The psychogenic theory holds that conversion symptoms (i.e., PNES) are not intentionally feigned but unconsciously generated [5], and symptom improvement rests on psychotherapy [6]. An estimated 15 to 30% of patients referred to epilepsy monitoring units (EMUs) for drug-resistant epilepsy (DRE) walk away with a diagnosis of PNES [7,8].

Experts have supported the approach of telling patients diagnosed with PNES that it was “good news” they did not have epilepsy [6]. This stance belies a presumption that the conversion disorder does not carry the same risks as epilepsy, notably, death associated with confirmed or suspected seizure activity, a phenomenon eponymously known as sudden unexpected death in epilepsy patients (SUDEP). The presumption of benign impact extends to prolonged seizures labeled PNES (i.e., pseudo-status epilepticus), which are common in this patient population [9,10,11]. Experts assert that these episodes are not dangerous or harmful to the patient, and thus should never be treated like prolonged and life-threatening epileptic seizures (i.e., status epilepticus) with benzodiazepines and anti-epileptic drugs (AEDs), which are considered unnecessary in pseudo-status and carry risk of iatrogenic harm [12].

Studies are now showing that patients with seizures labeled PNES are dying at an elevated rate comparable to patients with epilepsy, roughly three times above the general population [13,14,15]. Remarkably, both are dying from sudden unexpected death (SUD) associated with seizure activity [14,16,17], along with other natural and non-natural causes, including suicide [14,16,18].

The mortality rate reported in PNES patients akin to that of epilepsy has been deemed a “wake-up call” by PNES experts, with many chiming in to emphasize the importance of an accurate diagnosis [1,19] and the need for evidence-based treatments to reduce the morbidity and mortality in the PNES population [1]. Investigators point out the gravity of the conversion disorder and the failure of neurologists and psychiatrists to effectively treat these patients [20]. They assert that PNES patients are no less important than patients with epilepsy [19] and that the elevated mortality rate makes a strong case for treating PNES at least as purposefully and aggressively as epilepsy [21]. They have called for further studies to shed light on the recent findings [1] but the relief they seek is already available.

The purpose of this paper is to review the relevant clinical literature and present a hypothesis that seamlessly accounts for all of the empirical data, including the elevated mortality rates. The analysis has profound treatment ramifications for patients with scalp EEG-negative epileptic seizures (SNES).

## 2. Materials and Methods

The author completed an internet search of studies that investigated the mortality of PNES patients and compared those findings with the literature that addresses the mortality rate and causes of death in patients with epilepsy. To contextualize and appreciate this data, the analysis involved a further review of studies that describe the diagnostic practice in EMUs and that detail some noteworthy clinical observations in the PNES and epilepsy patient populations, including the natural history of these disorders and their response to epilepsy treatments.

## 3. Results and Discussion

Studies show that the standardized mortality ratios (SMR) in patients with epilepsy are 2 to 3 times higher than expected [15,22,23,24]. More than half of the fatalities are seizure related [25], and SUDEP represents a leading cause of death in patients with epilepsy [26]. SUDEP is defined as death in a patient with epilepsy that is not due to trauma, drowning, status epilepticus, or other known causes but for which there is often evidence of an associated seizure [24]. Patients not receiving any AEDs are at higher risk of SUDEP [27], and better disease control is associated with a decreased likelihood of associated sudden death [28]. The incidence of SUDEP is low among young children, more prevalent among adolescents, highest in young adults, and significantly decreased thereafter [27]. Additional causes of premature death in patients with epilepsy include suicide, AED effects, alcohol withdrawal, and aspirational pneumonia [29].

Status epilepticus (SE) is a medical emergency characterized either by continued seizures or by a lack of full recovery between seizures [30]. It is relatively common and associated with a mortality of approximately 20% [31]. Clinicians have long been urged to intervene early on, typically when the seizures have persisted beyond 5 min [32]. Delaying intervention can allow ongoing seizures to become refractory, with risk of neurologic harm and death, particularly from generalized tonic–clonic seizures [30]. Status epilepticus of all types is often associated with frontal lobe lesions [33,34], and complex partial status epilepticus, which can entail bizarre and apparently hysterical semiology, is common in frontal lobe epilepsy [35].

Drug-resistant epilepsy (DRE) is defined as the failure of adequate trials of two tolerated, appropriately chosen and used antiepileptic drug schedules (whether as monotherapies or in combination) to achieve sustained seizure freedom [36]. About 30 to 40% of patients with epilepsy end up developing DRE [8], and in one large study (N = 640), despite access to interventions, 61% of the subjects with DRE had ongoing seizures [37]. Once refractoriness is established, surgical treatment must be considered [38]. In select patients, epilepsy surgery is highly effective and leads to persistent improvements in the quality of life [38]. Accordingly, it is considered the standard of care for patients with DRE [38].

The provision of appropriate treatment for a medical condition depends on an accurate diagnosis and herein lies the crux of the matter. For decades, the scalp EEG test result has been hailed as the ‘gold standard’ for distinguishing a psychogenic from an epileptic seizure [39]. This, despite the fact that a significant percentage of epileptic seizures can only be detected with intracranial EEG electrodes. Studies have shown that simple partial seizures, complex partial seizures, and seizures with temporal and frontal lobe origins, can, and do fail to register on surface electrodes [8,35,40,41,42,43]. This limitation of the scalp EEG is not in dispute and PNES experts acknowledge that “the closest test to a biopsy for distinguishing epilepsy from PNES would be intracranial monitoring [43]”.

Intracranial EEG investigations confirm that SNES are receiving the erroneous label of PNES. Williamson [35] documented five case studies of patients with drug-resistant SNES of a frontal lobe origin who were misdiagnosed with the conversion disorder. The diagnostic error was recognized and corrected only after these patients were referred for intracranial monitoring, which proved the epileptic etiology of their SNES. Another group of investigators conducted an impromptu investigation using subdural strip electrodes in 12 patients diagnosed with PNES [17]. The intracranial monitoring proved that six of them suffered from DRE with complex partial seizures. Five were eligible for epilepsy surgery, and four achieved seizure freedom following that intervention. But for the impromptu investigation, these patients would have been referred for psychotherapy (for PNES) not epilepsy surgery. The remaining six patients demonstrated epileptiform spikes on the intracranial electrodes which cast some doubt on their PNES diagnoses. Though far less likely than their scalp counterpart, even intracranial electrodes can fail to capture epileptic seizures [17,40,44].

Experts assert that AEDs do not treat PNES [6] but the empirical data shows otherwise. The majority of PNES patients are treated initially for epilepsy with AEDs and often, for many years [45]. In a study that examined the delay to a diagnosis of PNES and the association with AED trials, investigators found that a positive response to AEDs was common in their subjects with lone PNES (N = 297) [45]. The observation that 30% of AED trials resulted in clinically significant reductions in PNES frequency was dismissed as a novel result [45]. In another retrospective study, 22 of 47 patients with lone PNES reported complete or partial remission of seizures on AEDs which was characterized as a placebo response by the PNES investigators [46]. In point of fact, AEDs do eliminate seizures labeled PNES, and they are more effective in this regard than the psychotherapy recommended by experts [47]. While the response of PNES to AEDs was patchy and limited in many of these patients, this same failure to achieve seizure-freedom on two or more AEDs is commonly found in the epilepsy patient population. It is called DRE.

Studies have shown that a highly effective intervention for eliminating seizures labeled PNES is no intervention at all. The spontaneous remission of PNES is well-documented [48,49] and mirrors the spontaneous remission of seizures in patients with untreated epilepsy [50,51,52]. In some patients, PNES stop right after the diagnosis is given [53], while in others, they simply remit with the passage of time [48]. The psychogenic theory cannot explain this remarkable parallel with epilepsy patients.

A review of the literature on PNES and epilepsy patient populations shows many other telltale similarities. The seizure semiology of PNES is “all too easily mistaken for epilepsy” and diagnostic error is “the rule rather than the exception [54]”. Both populations show pervasive brain disease, including structural alterations, and both are considered network disorders [55,56]. Epilepsy surgery has eliminated both seizure types in the same patient [57]. Traumatic brain injury is a risk factor for both disorders [58,59], and now, multiple studies are showing that they have a similar elevated mortality rate compared to the general population.

Duncan et al. [60] obtained death certificate information of a cohort of 260 patients who presented with PNES in Scotland between 1999 and 2004. Investigators found a significantly elevated rate of premature mortality (death before the age of 75) in their PNES subjects compared to the Scottish general population (0.58% versus 0.41%).

Jennum et al. [13] identified a cohort of 1057 patients receiving a first diagnosis of PNES from 2011 to 2016 in the Danish National Patient Registry and compared them to 2113 controls matched to age, sex, and geography. They found that the mortality rate in the PNES group was three times higher than controls.

Nightscales et al. [14] conducted a thorough retrospective cohort study of patients admitted for vEEG monitoring to three EMUs in Victoria, Australia. Three diagnostic groups were identified based on documented scalp EEG recordings: patients with PNES (N = 674), patients with epilepsy (N = 3067), and patients with both PNES and epilepsy (N = 176). Investigators found that the mortality in the PNES group did not differ significantly from the other two diagnostic groups and that the leading cause of death in the PNES subjects was “epilepsy” (N = 13, 23.6%). Two independent epileptologists reviewed those medical records and classified seven of these patients as having died from ‘definite or probable’ sudden unexpected death (SUD) and five, from ‘possible SUD’. The relative risk of mortality in the PNES group was increased by 8.6-fold in subjects younger than 30 and by 7.2-fold for those aged 30–39 years. The investigators concluded that patients diagnosed with PNES have a SMR 2.5 times above the general population and that they are dying at a rate comparable to patients with DRE.

The literature also contains anecdotal evidence that SUD is contributing to the elevated mortality rate in the PNES population. After a 15-year-old was referred for long-term scalp EEG monitoring, she was diagnosed with lone PNES and her AEDs were discontinued [17]. Three months later, she died of cardiac arrest during a witnessed seizure.

In the most recent study, Zhang et al. [18] identified subjects through multiple Swedish national registers with a diagnosis of PNES (N = 885), epilepsy (N = 50,663), and conversion disorder with motor symptoms or deficits (N = 1057), with 10 controls for each. The main outcome was all-cause mortality. The data showed that individuals with PNES had a 5.5 times higher risk of death compared to controls, and patients with epilepsy had a 6.7 times higher risk of death compared with individuals without epilepsy. The investigators concluded that like epilepsy, PNES carries a higher than expected risk of both natural and non-natural causes of death.

PNES experts have called for more studies to shed light on the unexpected findings [1] but the explanation is embedded in the existing data. The diagnostic practice of relying on the scalp EEG to distinguish an epileptic from a psychogenic seizure has led to diagnostic error in a substantial percentage of patients evaluated for DRE in EMUs. The large body of empirical data substantiates, that for decades, patients with SNES have been misdiagnosed by the ‘gold standard’ with PNES [61]. The reason the two patient populations mirror each other so closely is because they suffer from the same debilitating neurologic disorder, which is epilepsy.

## 4. Conclusions

The data showing an elevated mortality rate in the PNES population akin to epilepsy simply constitutes more confirmatory evidence that the former consists largely of patients with drug-resistant SNES. To reduce the morbidity and mortality in the PNES population, these patients must be given access to treatments for epilepsy including intracranial monitoring, AEDs and epilepsy surgery.

## Data Availability

Not applicable.

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
