# Peer review of "Psychogenic Nonepileptic Seizures—High Mortality Rate Is a ‘Wake-Up Call’"

_jpm, 2023, doi:10.3390/jpm13060892_

Round 1

Reviewer 1 Report

The author presented an interesting and important topic. Before acceptance, the author should better describe her methods and how she chose the articles in this paper.

The discussion is written poorly. Half of the results could be copied into discussion sections.

There is also no conclusion.

Author Response

I revised the Materials and Methods section. It is much more specific and I believe adequately addresses the reviewers critique. 

I agree with the reviewers position and recommended combining the Results with the Discussion section.  I also added a Conclusion. 

Reviewer 2 Report

I agree with the Autor in pointing up that the lack of EEG abnormalities do not play an important role in the differential diagnosis  in epileptology . During intra and concomitant extracranial recordings, infact, the epilectic activity can be very rich in the cortex but not all present in the scalp EEG. The contrary it is also true: some electrical epileptiform abnormality, without clinical manifestation, can not justify the diagnosis of epilepsy. In the Conclusion of the paper I suggest not to enphasize directly the intracranial studies but to deepen the diagnosis, recording the episodes clinically end with surface EEG together with a psicopatological study.

Author Response

I am a psychiatric expert and diagnostician. This article is about the fallacious diagnostic practice that is misdiagnosing epilepsy as a conversion disorder, which is a psychiatric diagnosis. I am not a neurologist and cannot 'deepen the diagnosis' as requested by the reviewer.